# A Review of *Pistacia lentiscus* Polyphenols: Chemical Diversity and Pharmacological Activities

**DOI:** 10.3390/plants12020279

**Published:** 2023-01-07

**Authors:** Chabha Sehaki, Nathalie Jullian, Fadila Ayati, Farida Fernane, Eric Gontier

**Affiliations:** 1BIOPI-UPJV Laboratory UMRT BioEcoAgro INRAE1158, SFR Condorcet FR CNRS 3417, UFR of Sciences, University of Picardie Jules Verne, 33 Rue Saint Leu, 80000 Amiens, France; 2Laboratory of Natural Resources, University Mouloud Mammeri of Tizi-Ouzou, Tizi Ouzou 15000, Algeria

**Keywords:** *Pistacia lentiscus*, polyphenols, phenolic acids, flavonoids, yields, pharmacological properties

## Abstract

*Pistacia lentiscus* (lentisk) is a plant species of the Anacardiaceae family. It is a medicinal plant that grows wild in the Mediterranean region. This review aims to update the existing knowledge regarding *P. lentiscus* polyphenols by consulting references dated from 1996 to 2022. The data are organized and analyzed as follows: (i) to show the chemical diversity of phenolic products from *P. lentiscus*; (ii) to summarize the variability in phenolic composition and quantity; this could be attributed to plant origin, environmental conditions, phenological stage, and the polarity of the extraction solvents; (iii) to present the pharmacological properties in agreement with the traditional uses of this plant; and (iv) to demonstrate the correlation between the chemical profile and the pharmacological effect. Various compositions were observed, including phenolic acids, flavonoid glycosides, anthocyanins, catechins, and their derivatives. The biological and therapeutic potentials of lentisk extracts have been evaluated in terms of antioxidant, antimicrobial, and anti-inflammatory activities. Most of these activities are related to the phenolic composition of this plant. The content of this review will undoubtedly contribute to the choice of techniques for isolating the different bioactive molecules contained in the *P. lentiscus*. It is also of significance for the potential development of a micro-industrial sector based on the valorization of lentisk polyphenols.

## 1. Introduction

*P. lentiscus* L. is an evergreen shrub of the Anacardiaceae family that is widely distributed in Mediterranean countries [1]. This plant is also called lentisk [2]. It has been used in traditional medicine for the treatment of several diseases, such as for gastrointestinal diseases, eczema, and throat infections [1,2], due to its potent antioxidant, anti-inflammatory, and antimicrobial effects [3]. The aerial part helps in the treatment of hypertension and has diuretic properties [4]. It also acts as an antioxidant [5], antiproliferative [6], and antimicrobial agent [7]. It has been reported to be effective in the treatment of scabies and rheumatism [8]. Its different parts contain a variety of chemical compounds that are medically important such as resin, essential oils, gallic acid, anthocyanins, flavonol glycosides, triterpenoids, tocopherol, and arabinogalactan proteins [9]. These health-promoting properties of lentisk have also been attributed to the presence of various biologically active compounds (BACs) such as phenolic compounds [10].

Polyphenols represent a class of the most well-known secondary plant metabolites [11]. This category of compounds includes more than 8000 molecules [12]. These compounds contain one or more aromatic rings linked to one or more hydroxyl groups in their basic structure and are divided into different classes, such as phenolic acids, flavonoids, tannins, stilbenes, and lignans [13]. Polyphenols are known for their health-promoting effects, such as their antioxidant and anti-inflammatory activities [14,15]. Polyphenols contribute to decreasing the incidence of cardiovascular diseases [16], liver disorders [17], obesity [18,19], and colon cancer [20]. These compounds have received more attention in recent years, and many aspects of their chemical and biological activities have been identified and evaluated [21,22].

Many studies have been carried out regarding the identification and quantification of the chemical composition of *P. lentiscus* secondary metabolites as well as their biological and pharmacological properties. However, there are no reviews summarizing the detailed investigations that have been carried out on lentisk polyphenols. Therefore, this review aims to (i) illustrate the composition and content of phenolic compounds in the different parts of lentisk; (ii) examine the variation of its composition and polyphenol yield under the influence of geographical origin, the part of the plant used, and the type of extraction solvents; and (iii) illustrate the pharmacological properties of the different phenolic extracts. The different parts of this review are linked to establish a scientific basis for the curative capacities of this plant by explaining the relationship between the phytochemistry and pharmacology of lentisk phenolic compounds.

## 2. Research Methodology

To prepare a review article on the chemical diversity and pharmacological activities of polyphenols from the plant *P. lentiscus*, existing knowledge was integrated. The corresponding data were collected by reviewing articles dating from 1996 to 2022 published in Elsevier, Google Scholar, MDPI, Taylor and Francis, and Springer databases.

## 3. Quantitative Analyses of *P. lentiscus* Phenolic Classes

Polyphenols are reactive metabolites abundant in plant-derived foods, particularly fruits, seeds, and leaves. Polyphenolic compounds are specialized plant metabolites found in numerous plant species. Recent phytochemical analyses have shown that all the parts (leaf, stem, fruit, and root) of *P. lentiscus* are rich in bioactive phenolic components. The results of these analyses are shown in Table 1. Lentisk extracts were found to be composed of total polyphenols, flavonoids, flavonols, tannins, anthocyanins, and proanthocyanidins. Depending on the type of extracts, as well as the part of the plant used, one can observe a variation in the yield of these metabolites. Classically, phytochemical analyses were based on the use of specific colorimetric methods for each chemical family. Their contents were determined by spectrophotometry. The total phenolic content was evaluated using the Folin–Ciocâlteu reagent with gallic acid (GA) as standard. The flavonoid content was determined using the aluminum tritrichloride method with quercetin (Q) as standard. The total tannin (TA) content was estimated based on their precipitation using bovine serum albumin (BSA) protein. The concentration of proanthocyanidins was determined using a butanol-HCl test (Table 1).

### Variability in Phenolic Classes in P. lentiscus

Variable levels of total phenolic contents, flavonoids, and condensed tannins were found in all parts of *P. lentiscus.* According to the results shown in Table 1, the total phenolic content varied from 17 mg gallic acid equivalent/g dry matter (DM) to 955 mg gallic acid equivalent/g DM in the *P. lentiscus* extracts [32,38]. For the other polyphenolic classes, the low concentration of flavonoids in the lentisk extracts was 0.9 µg quercetin equivalent/g DM and the higher concentration was 278 mg rutin equivalent/g DM [34,36]. For the contents of condensed tannins, according to the Table 1, the minimum value was 7 mg catechin equivalent/g DM and the maximum value was 997 mg tannic acid equivalent/g DM [24,29]. Other compounds were also determined in lentisk extract, such as myricetin-rhamnoside (3.6 mg/g DM) and quercetin-rhamnoside, (1.6 mg/g DM) [33]. Flavonols (7.2 mg quercetin eq/g DM) and proanthocyanidins (39.2 mg catechin equivalent/g DM) were also detected [30].

The data gathered in Table 1 show that the amount and nature of polyphenolic compounds in the different lentisk extracts largely depend on the geographical origin, the sampling period [35,39], the plant part [29,37,38], and the type of solvent used during the extraction [26,27,31].

According of the plant part, the total phenolic content in the leaves was significantly higher than that revealed in the stems and fruits in most of the studies described in Table 1. Root extracts have a relatively low content of phenolic compounds [29]. The highest concentrations of flavonoids and condensed tannins were recorded in the leaves and stems of *P. lentiscus*.

The solvent used during extraction plays a key role in the variation in the amount of polyphenols [26,27,31]. The determination of the amount of polyphenols for the same plant part extracted with different solvents revealed quantitative differences in the lentisk phenolic contents. Studies have shown that the total phenolic content and flavonoids in the methanolic extract from leaves were higher than those in the aqueous extracts [31]. This result is in agreement with previously studies [29,35,39,40,41] that revealed that methanol is an interesting and suitable solvent for extracting bioactive chemicals from plants, such as phenolic and flavonoid components. Other works aimed to quantify the polyphenolic content according to the solvent used for extraction. Salhi et al. (2019) [42] prepared extracts using dicloromethane, ethyl acetate, methanol, ethanol, or water as solvents. The results of this study, listed above in Table 1, illustrate that the total phenols, total flavonoids, and total flavonols decreased in the following order according to the solvent extracts used: ethanol > water > methanol > ethylacetate > dichloromethane. The results obtained also indicate that amongst the five extracts, the ethanol extract contained the highest amount. The quantitative analyses performed prove that the whole plant contains a high amount of total phenols. This amount increases as the polarity of the extraction solvents used increases. Other factors can impact the polyphenolic content in lentisk, such as the sampling period of the plant [35,38]. Yosr et al. (2018) [38] studied the variation in polyphenol content in acetone extracts of different lentisk parts during four vegetative stages of female and male shrubs. For all the growth cycle periods of the plants, the male vegetative organs had significantly higher phenolic contents than the females. In both sexes, the leaves showed the maximum contents in contrast to the stems, flowers, and fruits. Flavonoid and tannin contents were also consistently higher in the males than in the females for all the tested organs. Flavonoids in leaf and flower extracts were higher than those in fruits and stems. The tannin content was in the following order (highest to lowest): leaves > stems > flowers > fruits.

The total phenolic and flavonoid contents in the vegetative organs of both genders varied between the harvesting times. A decrease in the content of compounds depending on plant part was observed from the vegetative dormancy stage (December) to the late ripening stage [38].

Therefore, we can state that the great variability in phenolic composition and quantity could be attributed to plant origin, environmental conditions, phenological stage, extraction methods, and the polarity of the organic solvents used.

These studies showed that leaf extracts of *P. lentiscus* contained high levels of total phenolic compounds. Therefore, the results obtained are useful for further research such as in the identification of specific phenolic compounds. Analytical methods for the determination of the composition have been developed. We discuss this point in more detail in the next section.

## 4. Identification and Quantification of Individual Phenolic Compounds

The separation, identification, and quantification of polyphenols were performed on several extracts of *P. lentiscus* from Mediterranean countries. Several chromatographic techniques, including semi-preparative high-performance liquid chromatography (HPLC) and HPLC–photodiode array detection by comparison with authentic standards. High-performance liquid chromatography (HPLC)–photodiode array detection (DAD)–mass spectrometry (MS) (HPLC-MS) equipped with an electrospray ionization (ESI) interface and ^1^H and ^13^C NMR was used. These techniques allowed us to obtain qualitative and quantitative information on the biochemical constitution of *P. lentiscus*. Several metabolites including flavonoids, flavonols, flavanols, flavones, flavonoid glycosides, myricetin derivatives, anthocyanins, catechins, and different phenolic acids and their derivatives were identified (Table 2).

Several phenolic acids were detected, among which gallic acid was recovered by several authors [44,46,49,56,57]. Quinic acid derivatives and galloyl derivatives of both glucose and quinic acid, such as 3,5-O-digalloyl quinic acid and 3,4,5-tri-O-galloyl quinic acid (Figure 1), beta-glucogallin, were detected [46,57,58,59]. Some other studies also reported the presence of tannic acid, chlorogenic acid, caffeic acid, vanillic acid, p-coumaric acid [60], ferulic acid, 3,4-dihydroxybenzoic acid, and trans-cinnamic acid [44,60].

The analysis of the different parts of the *P. lentiscus* extracts using UHPLC-ESI-MS identified different metabolites including flavonoid glycosides, anthocyanins, and catechins and their derivatives. Among flavonoids, flavonol glycosides were found to be the most abundant class of flavonoids in the leaves and fruits of *P. lentiscus* [57]. Myricetin-3-O-rhamnoside (Figure 1) was found to be the predominant flavonol glycoside, followed by myricetin-3-O-glucoside, myricetin-3-O-glucuronide, and quercetin-3-O-rhamnoside [1]. Myricetin derivatives were also determined to account for 20% of the total amount of polyphenols in *P. lentiscus* leaves [47,49,57]. Bozorgi et al. (2013) [59] reported that quercetin-3-glucoside is the major flavonol in *P. lentiscus* as well as in other *Pistacia* species. Flavanols assigned as procyanidin B1, procyanidin B2, epicatechin, catechin, epigallocatechin gallate, and epicatechin gallate were recovered in several studies. Catechin was the most representative flavanol in *P. lentiscus* leaves [29,44], while procyanidin B1 (Figure 1) was the most abundant flavanol in fruit extracts [57]. Belonging to the class of flavones, luteolin and apigenin were previously reported by Bampouli et al. (2014) [10] in *P. lentiscus* leaves. It can be observed that the concentration of luteolin was significantly higher than apigenin in both leaves and fruit extracts. Mehenni et al. (2016) [44] found luteolin to be the second most abundant polyphenol in fruit. Anthocyanins, namely, delphinidin-3-O-glucoside, cyanidin-3-O-glucoside [49], and cyanidin 3-O-arabinoside [50] were detected in *P. lentiscus* berries and leaves. The major anthocyanins in *P. lentiscus* berries were identified to be cyanidin-3-O-glucoside (Figure 1). Delphinidin-3-O-glucoside and cyanidin 3-O-arabinoside were also found but in minor quantities [50].

Table 2 completes Table 1 by identifying the phenolic molecules belonging to the different polyphenolic classes noted in Table 1. The data presented in Table 2 also show the variability in the nature of polyphenols in lentisk under the influence of the factors described above (the part of the plant used, the extraction method, and the extracting solvent and geographical origin). It is reported that lentisk leaves are a rich source of polyphenolic compounds (5–7% of the dry matter of the leaf), especially galloyl derivatives such as mono-, di-, and tri-O-galloyl quinic acid and mono-galloyl glucose [49]. Siano et al. (2020) [53] reported the presence of other phenolic acids such as vanillic and syringic acid derivatives, hydroxycinnamic acid, and flavonoids in lentisk seed oil extracted from fruits. The digalloyl quinic acid derivative was detected as the main phenolic component in the three parts of the plant (leaf, fruit, and stem), at percentages of 61.38%, 13.5%, and 45.11% in the stem, fruit, and leaf, respectively [60]. Dragović et al. (2020) studied the effect of the extraction solvent (ethanol or methanol) on the polyphenol composition of *P. lentiscus* leaf extracts from four locations at three different phenological stages. In both ethanolic and methanolic leaf extracts, phenolic acids and flavonol glycosides were quantified, whereas catechin was quantified only in methanolic extracts. The mass concentration of these compounds varied significantly according to phenological stages, cultivation sites, and extraction solvents used [1].

## 5. Prominent Pharmacological Activities

The plant *P. lentiscus*, which is used in traditional medicine, possesses pharmacological attributes and may offer significant potential as a therapeutic agent [28,29,30,34,42,43,51,53]. As shown in Table 3, several studies have therefore been undertaken to justify its traditional use and to establish a scientific basis for the curative capacities of this plant. The most marked pharmacological properties are its antioxidant [28,29,30,43], antimicrobial [28,34,42], anticancer [51,53], and anti-inflammatory [31,43] activities. The phytochemistry of the lentisk extracts tested for the curative properties described above is also discussed in these same studies (Table 1 and Table 2). We are interested in summarizing and highlighting the correlation between the chemical composition and the pharmacological effect of this plant, particularly for its antioxidant, anti-inflammatory, antimicrobial, and anticancer properties. To explain this relationship, we drew on data based on the phytochemical compositions presented in Table 1 and Table 2 and the pharmacological effects illustrated in Table 3, and also on other studies in the literature that aimed at explaining the link between the phytochemistry and pharmacology of the plant.

### 5.1. Antioxidant Activity

Different methods were used to evaluate the antioxidant capacity of phenolic extracts of *P. lentiscus*: DPPH (2,2-diphenyl1-picrylhydrazyl), i.e., free radical scavenging activity; free hydroxyl radicals (HO); ferric-reducing power (FRAP); the carotene bleaching (CB) assay; the oxygen radical absorbance capacity (ORAC) test; hydrogen peroxide scavenging activity (H_2_O_2_); the phosphomolybdenum (TAC) assay; and the ABTS·+: 2,2′-azino-bis-(3 ethylbenzthiazoline-6-sulphonic acid) assay were the most used (Table 3). These methods have different modes of action. Electron transfer, proton transfer, and iron reduction are the main mechanisms involved. The observations noted after the evaluation of the antioxidant potential of this plant are illustrated in Table 3. Investigations (Table 3) were conducted regarding the evaluation of antioxidant activity by comparing to reference antioxidants (ascorbic acid, gallic acid, trolox, rutin, and quercetin). The results of these investigations revealed that lentisk’s antioxidant properties are comparable to those of the reference antioxidants. The extracts of *P. lentiscus* remain a considerable source of natural antioxidants.

The ability of the *P. lentiscus* extracts to prevent free radicals could be attributed to the high content of phenolic compounds [23,29,30,34]. There is a significant positive correlation between the antioxidant test results and the amount of total phenols [37]. The antioxidant capacity of *P. lentiscus* extract has been shown to be due primarily to gallic acids and their galloyl derivatives (5-Ogalloyl; 3,5-O-digalloyl; 3,4,5-tri-O-galloyl) [23]. The trapping of DPPH increases accordingly with the number of galloyl groups [23]. Quercetin and gallic acid are powerful natural antioxidants [23], and monophenols are less effective than polyphenols [61,62]. With gallic acid, the inductive effect of these three hydroxyl groups is a significant factor influencing the increase in antioxidant activity [61]. This suggests that they are partly responsible for the antiradical potential observed in the lentisk extracts [43]. In addition, the antioxidant activity also depends on the polarity of the solvent, the solubility of the phenolic compounds, and the hydrophobic nature of the reaction medium [42]. A high polarity of the extract solvent increases the antioxidant capacity (DPPH and FRAP) of the lentisk extracts. Polar extracts such as ethanol, methanol, and aqueous extracts were found to be rich in polar compounds, which are either hydrogen atom donors or singular atom transfer agents [42]. In addition to the solvent polarity, the solubility of the phenolic compounds is governed by the degree of polymerization of the phenols, the part of the plant used, and the variability of soil and climatic conditions [63,64]. The flavonol glycosides found in lentisk [1,43,57] are known for their antioxidant attributes, which are strongly related to their structural characteristics, i.e., the hydrogen donor substituents (OH groups) and the presence of a 2,3 double bond that increases their scavenging capacity and inhibition of pro-oxidant enzymes [24]. It has been shown that the main compound present in lentisk extracts, i.e., myricetin-rhamnoside [1,43], has a DPPH scavenging capacity comparable to that of vitamin C [65].

**Table 3 plants-12-00279-t003:** Pharmacological activities of lentisk phenolic extracts.

Pharmacological Activity	Plant Part	Product	Method	Significant Results)	Ref
**Antioxidant activity**	Leaves	Ethanolic extract from leaves	DPPH (2,2-diphenyl- 1-picrylhydrazyl): free radical scavenging activity	The extract (3 g/L) inhibited 95.69% of the activity of the DPPH radicals	[23]
	Leaves	Dichloromethane extract Ethylacetate extract Ethanol extract Methanol extract Aqueous extract	DPPH free radical scavenging activity Ferric-reducing activity power (FRAP) Carotene bleaching (CB) assay	IC50 = 05.44 (μg/mL) (Ethanolic extract) 309.60 mg ascorbic acid equivalent/g extract (methanolic extract) Inhibition = 90.32% per 2 g/L of extract (dichloromethane extract)	[42]
	Leaves	Ethanolic extract	Oxygen radicalabsorbance capacity(ORAC) test	Antioxydant capacity = 5865 µmol TE/100 gE	[43]
	Leaves	Aqueous extract	Free radical scavenging activity (DPPH assay) Hydrogen peroxide scavenging activity (H_2_O_2_) Ferric-reducing power (FRAP) assay Antioxidant assay by phosphomolybdate method	IC50 (aqueous extract) = 9.86 μg/mL	[28]
	Berries	Ethanolic extract	DPPH assay (ABTS·+: 2,2′-azino-bis-(3 ethylbenzthiazoline-6-sulphonic acid)) assay Reducing power activity assay	IC50 = 8.60 mg/mL IC50 = 8.65 mg/mL IC50 = 12.21 mg/L	[32]
	Leaves	Methanolic extract	DPPH assay β-carotene bleaching test	IC50 = 0.008 mg/mL IC50 = 0.12 mg/mL	[30]
	Leaves	Aqueous fraction obtained from chloroformic extract	Reducing power assay Scavenging ability against DPPH radical Activity against linoleic acid peroxidation	IC50 = 50.03 lg/mL IC50 = 4.24 lg/mL IC50 = 0.82 lg/mL	[33]
	Leaves	Methanolic fraction from chlorformic extract	DPPH assay ABTS assay	inhibition of 50% of DPPH radicals (2.9 μg/mL extract) inhibition of 50% of ABTS•+ (0.6 μg/mL extract)	[51]
	Fruits	Aqueous extract Ethyl acetate extract Butanol extract	Free radical scavenging (DPPH assay)	100 mg/mL of aqueous extract inhibited 86.13% of DPPH radicals	[66]
	Leaves	Ethyl acetate fraction from ethanolic extract	Ferric-reducing power assay (FRAP)	IC50 = 15.0 μg/mL (FRAP)	[34]
	Fruits	Phenolic extract from vegetable oil	DPPH assay	Significant antioxidant power (IC50 = 37.38 mg/mL)	[36]
	Fruits Twigs Leaves	Aqeuous extract Hexane extract Ethyl acetate extract Methanol extract Ethanol extract	Phosphomolybdenum (TAC) assay	The aqueous extract of *P. lentiscus* leaves showed the highest TAC with 488.16 mg AA/g of extract.	[37]
	Leaves Fruits	Methanolic extracts	Free radical DPPH assay	EC 50 = 0.121 mg/mL for leaves and EC 50 = 0.26 mg/mL for fruits	[67]
	Aerial parts	Methanolic extract	FRAP	reducing power = 84.6–131.4 mmol Fe2+/L plant extract	[35]
**Antibacterial activity**	Leaves	Dichloromethane extract Ethylacetate extract Ethanolic extract Methanolic extract Aqueous extract	The disk diffusion method on Muller–Hinton agar (MHA).	All these extracts had efficient antimicrobial activity against: Gram-positive bacteria: *Micrococcus luteus*, *bacillus subtilis*, and *listeria innocua* Gram-negative bacteria: *Escherichia coli* The activity was almost the same for all the extracts against each bacterium	[42]
	Leaves	Aqueous extract	The disc diffusion method	A maximum inhibition zone of 12 mm was observed on *Pseudomonas aeruginosae*, while moderate activity was obtained against all strains	[28]
	Leaves	Decoction Petroleum ether extract Ethanol extract Maceration Infusion	The minimal inhibitory concentration (MIC) was determined by a microdilution assay in microtiter plates The minimal bactericidal concentration (MBC) was determined by carrying out a subculture of the tubes showing no growth on plates	Antimicrobial activity against: *Staphylococcus aureus* and *Escherichia coli* Decoction showed the best activity (MIC = 312 mg/L for all the three bacterial strains). MIC and MBC values were the same, so the substances should possess bactericidal activity	[68]
	Leaves	Ethyl acetate fraction from ethanolic extract	The MIC of the extract was determined using the agar dilution method The MBC was determined by taking samples from the nutrient agar plates that showed no visible growth after 24 h incubation and subculturing them in tubes containing nutrient broth	Moderate inhibitory activities against: *Staphylococcus aureus*, *Listeria innocua*, *Bacillus cereus*, *Escherichia coli*, *Salmonella typhi*, *Salmonella enterica*, *Pseudomonas aeruginosa*, *Proteus mirabilis*, *Vibrio cholerae*, and *Enterococcus faecalis* There was remarkable activityagainst *Vibrio cholerae* with an MBC value of 0.3 mg/mL	[34]
	Leaves	Methanolic extract Aqueous extract	The disk diffusion method	Antibacterial activity against: *Staphylococcus aureus*, *Staphylococcus haemolyticus*, *Pseudomonas aeruginosa*, and *Proteus mirabilis* Methanol extract showed a significant inhibitory effect on the growth of all tested bacterial isolates, with 33 mm and 27 mm against *S. aurous* and *S. haemolyticus*, respectively	[69]
**Antifungal activity**	Leaves	Dichloromethane extract Ethylacetate extract Ethanolic extract Methanolic extract Aqueous extract	The disk diffusion method on Muller–Hinton agar (MHA).	Significant antifungal activity against: *Candida pelliculosa* and *fusarium oxysporum albidinis* The ethanolic extract was the most active	[42]
	Leaves	Hydro-methanolic extract (70/30 *v*/*v*)	Diffusion using solid medium method	The extract was more active against *Trichophyton mentagrophyte* and *Microsporum canis*, with growth inhibition: *Trichophyton mentagrophyte* (17 mm) *Microsporum canis* (16.7 mm)	[70]
	Leaves	Aqueous extract	The minimal inhibitory concentration (MIC) was determined by a microdilution assay in microtiter plates	Antifungal activity against: *Candida albicans*, *Candida parapsilosis* and *Cryptococcus neoformans* The highest activity of *P.lentiscus* was against *T. glabrata* (MIC = 39–156 mg/L)	[68]
	Leaves	Ethyl acetate fraction from ethanolic extract	The MIC of the extract was determined using the agar dilution method	Good antifungal activity against *Candida albicans* with CMI 0.1 mg/mL	[34]
**Anticancer activity**	Leaves	Ethanolic extract	The in vitro cytotoxicity of the extract was determined by sulforhodamine B (SRB) assay	Moderate cytotoxic activity against lung cancer A549, breast cancer MCF7, prostate cancer PC3, and HepG2 liver cancer	[45]
	Leaves	Ethanolic extract	3-(4,5-dimethylthiazol-2-yl)-2,5- diphenyltetrazolim bromide (MTT) assay	Anticancer potential against melanoma (B16F10) cell lines	[43]
	Leaves	Methanolic fraction from chlorformic extract (sonication)	MTT, SRB, and LDH assays forSH-5YSY, and SK-N-BE(2)-C human, and neuronal cell lines, and also on C6 mouse glial cell line	Significant cytoprotective response in both the oxidized cell systems	[51]
	Edible fixed oil (fruits)	Hydro-methanolic extract (methanol 80%, *v*/*v*)	A crystal violet viability assay with increasing concentrations was carried out	The extract induced clear dose-dependent effects on the growth of the HT-29 cell line derived from human colorectal adenocarcinoma	[53]
	Leaves	Hydro-methanolic extract (8:2 *v*/*v*)	Cell viability by MTT assay	The extract showed activity on: the SK-N-BE(2)C cell line with an IC50 value of 100.4 ± 1.6 μg/mL the SH-SY5Y cell line with IC50 value of 56.4 ± 1.1 μg/mL	[60]
**Anti-inflammatory activity**	Leaves	Ethanolic extract	The measurement of the secretion of interleukin-1 by macrophages exposed to ATP or H_2_O_2_ on the THP-1 monocytic cell line	Significant anti-inflammatory activity	[42]
	Leaves	Methanolic extract	Albumin denaturation inhibition method in human red blood cell suspension	Apparent anti-inflammatory activity	[46]
	Leaves	Chloroformic extract Ethyl acetate extract Methanolic extract	The carrageenan-induced paw edema assay	MeOH extract presented the best anti-inflammatory activity Dose of 200 mg/kg showed 68% edema inhibition	[71]
	Leaves	Methanolic extract (maceration) Aqueous extract (decotion)	Three inflammation models: Croton oil-induced ear edema in mice Carrageenan induced-pleurisy in rats Acetic acid-induced vascular permeability in mice	Local treatment with 2 mg/ear of: alcoholic extract significantly decreased ear edema (65%) aqueous extract exerted a lower inhibitory effect (51%). methanolicand aqueous extracts: at (400 mg/kg) inhibited neutrophil migration by 29% and 38%, respectively; at methanolic and aqueous extracts (100 μg/mL) inhibited neutrophil chemataxis by 81% and 71%, respectively	[31]
	Fruits	Acetonic extract	The ear edema model induced by Croton oil and the airpouche model induced by lambda carrageenan	Oral administration dose of 300 mg/Kg of extract decreased ear edema by 80% Dose of 1 mg of extract/pouche decreased pouch edema by 34%	[25]

MTT: 3-(4,5-dimethyl-2-thiazolyl)-2,5-diphenyl-2Htetrazoliumbromide. SH-SY5Y: neuroblastoma cells were purchased. SK-N-BE(2)-C: human bone marrow neuroblastoma cells. SRB: sulforhodamine B.

### 5.2. Antimicrobial Activity

Researchers evaluated the antimicrobial and antifungal activities of lentisk phenolic extracts (Table 3). The evaluation of the antimicrobial activity was based on the comparison of the antimicrobial effects of lentisk extracts and those of the different antibiotics (such as amphotericin, amoxicillin, and ciprofloxacin) used in the studies presented in Table 3. The results of these studies revealed that the extracts from leaves show an interesting antimicrobial potential against Gram-positive and Gram-negative bacteria, but also against different fungi. The disk diffusion method and/or the dilution method were the most useful methods. Lentisk leaf extracts prepared in different solvents (dichlotomethane, methanol, ethanol, ethyl acetate, and water) were all found to be effective against the bacteria *Micrococcus luteus*, *Bacillus subtilis*, *Listeria innocua*, and *Escherichia coli* [42]. The same extracts also have an inhibiting effect against fungal strains as *Candida pelliculosa* and *Fusarium oxysporum albidini* [42]. Lauk et al. (1996) [68] showed that lentisk leaf decoctions have good antibacterial activity against *Sarcina lutea*, *Staphylococcus aureus,* and *Escherichia coli* (with MIC = 312 mg/L for the three bacteria tested). The activity against fungal cells, *Candida albicans*, *Candida parapsilosis*, *Torulopsis glabrata*, and *Cryptococcus neoformans* seems to be much more interesting [64]. Bakli et al. (2020) [34] showed that the ethyl acetate fraction from an ethanolic leaf extract had remarkable activity against *V. cholerae* with a value of MBC 0.3 mg/mL [34]. Another study showed that methanol extracts had a significant inhibitory effect on the growth of bacterial isolates (*Staphylococcus aureus*, *Staphylococcus haemolyticus*, *Pseudomonas aeruginosa*, and *Proteus mirabilis*). This methanolic extract showed a higher zone of inhibition than that of the aqueous extract. The maximum zone of inhibition was observed in the methanolic extract at 100% concentration with 33 mm and 27 mm against *S. aureus* and *S. haemolyticus*, respectively [69]. *P. aeruginosa* was of particular interest, as this bacterium was inhibited by the methanol extract to a greater extent than with the use of the reference antibiotic levofloxacin [69]. In most of these studies, Gram-negative bacteria were found to be less sensitive to the extracts than Gram-positive bacteria.

This can be explained due to the presence of hydrophobic lipopolysaccharides in the outer membrane, which provide protection against different agents [72], in addition to enzymes in the periplasm that destroy foreign molecules introduced from the outside [73]. The antimicrobial activity observed in this study could be due to the presence of flavonoid compounds (kaempferol, myricetin, quercetin, and rutin) in the lentisk extracts (see Table 2). Flavonoids are known for their antimicrobial activity against a wide range of microorganisms [34]. They have multiple cellular targets and can be applied to different components and functions in the bacterial cell [74,75]. These kinds of molecules promote antimicrobial activity against human pathogenic microorganisms [76].

### 5.3. Anticancer Potential

Some studies (Table 3) aimed to evaluate the anticancer potential of different phenolic extracts of this plant in vitro. The ethanolic leaf extracts showed anticancer potential against lung cancer A549, breast cancer MCF7, prostate cancer PC3, and HepG2 liver cancer in vitro [45]. Yemmen et al. (2017) [60] revealed the anti-proliferative activities of leaf hydro-methanolic extracts against two human neuroblastoma cell lines (SK-N-BE(2) C and SH-SY5Y) with IC50 values of 100 μg/mL and 56 μg/mL, respectively. The anticancer potential of the crude extracts against melanoma (B16F10) and breast (EMT6) cell lines was also evaluated. The leaf and fruit extracts inhibited the growth of B16F10 cells (IC50 = 56 and 58 μg/mL, respectively) [42].

Phytochemical studies have indicated the presence of significant amounts of flavonoids, tannins, and phenolic compounds in *P. lentiscus* extracts (see Table 1 and Table 2) [24], which may be responsible for the anticancer activity of lentisk extracts. It has been shown that the presence of a 2,3-double bond and three adjacent hydroxyl groups in the structure could confer a higher anticancer potential to a flavonoid [77]. One example of the flavnoids found in lentisk extracts is myricetin (see Table 1 and Table 2); this molecule was found to have significant cytotoxic activity against B16F10 melanoma cell cultures [77].

### 5.4. Anti-Inflammatory Activity

*P. lentiscus* is used in traditional medicine for the treatment of inflammation, burns, and gastrointestinal disorders. Anti-inflammatory activity has been the focus of many recent investigations (Table 3). Dellai et al. (2013) [71] examined the efficacy of aqueous and organic extracts of *P. lentiscus* leaves in vivo for their anti-inflammatory and anti-ulcerogenic activities using the carrageenan-induced paw edema assay and HCl/ethanol-induced gastric injury in rats, respectively. Aqueous (AQ), chloroformic (CHCl_3_), ethyl acetate (EtOAc), and methanolic (MeOH) leaf extracts administered intraperitoneally showed a dose-dependent anti-inflammatory effect. Leaf extracts of CHCl_3_, EtOAc, and MeOH, administered orally, showed concentration-dependent inhibition of gastric lesions. The effect of all the extracts in both activities is comparable to the reference drugs: cimetidine and acetylsalicylate of lysine, respectively [71]. The pharmacological evaluation of the *P. lentiscus* leaf extracts showed the anti-inflammatory potential of this plant and that its activity is unlike non-steroidal anti-inflammatory drugs and corticosteroids. The extracts did not cause damage to the stomach mucosa but showed an inhibition of lesion formation. The work carried out by Bouriche et al. (2016) [31] concerns the measurement of the anti-inflammatory activity of alcoholic and aqueous extracts of *P. lentiscus* leaves. Croton oil-induced ear edema in mice was used as a model of acute inflammation. The results showed that a local treatment with 2 mg/ear of alcoholic extract significantly decreased ear edema (65%), while the aqueous extract exerted a weaker inhibitory effect (51%). The anti-inflammatory activity of lentisk was otherwise examined by measuring the secretion of interleukin-1β by macrophages exposed to ATP or H_2_O_2_. The leaf extract (100 μg/mL) showed significant anti-inflammatory activity compared to acetylsalicylic acid (ASA) [31].

Constituents such as quercetin derivatives (quercetin 3-O-rhamnoside, see Table 2) [44] participate in the promotion of anti-inflammatory effects such as the attenuation of swelling [44]. This has been proven in mice [78].

The study conducted by Bouriche et al. (2016) [31] (Table 1 and Table 3) proved that the anti-inflammatory effect of the acetone extract of *P. lentiscus* fruit was similar to that of indomethacin, which is used as a standard anti-inflammatory agent. The mechanism of action of indomethacin on inflammation is based on the inhibition of pro-inflammatory prostaglandin synthesis [79]. The anti-inflammatory effect of the acetone extract of *P. lentiscus* fruit is probably attributed to the lipophilic soluble substances that are able to penetrate through the skin barrier [80] and can thereby exert their anti-inflammatory effects. Likely candidates for these anti-inflammatory substances are flavonoids and polyphenols, which have been isolated from *P. lentiscus*. Phenolic compounds are known to interact with and penetrate through lipid bilayers [81]. The observed anti-inflammatory effect is also likely due to the presence of antioxidant compounds in the extract.

The anti-inflammatory results [25,31,43,46,71] provide valuable evidence regarding the anti-inflammatory potential of *P. lentiscus* leaves, suggesting that this plant can be exploited as a natural source of anti-inflammatory agents.

The results obtained in all the studies illustrated above indicate that the extracts of *P. lentiscus* present antioxidant, anti-inflammatory, anticancer, and antimicrobial properties in agreement with the traditional uses of the plant.

## 6. Other Activities

### 6.1. Antidiabetic Activity

The antidiabetic potential of mastic preparations (decoctions and infusions) is already known among people in rural areas. Scientific investigations have been carried out on the evaluation of the antidiabetic activity of lentisk extracts in vitro and in vivo by measuring the inhibition of α-amylase and α-glucosidase enzymes (digestive enzyme that hydrolyzes polysaccharides). The results of the study carried out by Mehenni et al. (2016) [44] showed that the leaf and fruit ethanolic extracts possessed promising antidiabetic activity in streptozotocin-induced diabetic rats, similar to that of the reference drug glibenclamide (0.91 g/L). This result was confirmed by in vitro tests for the inhibition of α-amylase [44]. Further investigations confirmed that aqueous extracts of the leaves were identified as strong and effective dual inhibitors of α-amylase and glucosidase in vitro [52].

### 6.2. Hepatoprotective Activity

The hepatoprotective effect of lentisk extracts has also been reported. Mehenni et al. (2016) [44] performed a histological examination of the liver of mice pretreated with the same dose of leaf and fruit extract (125 mg/kg body weight) against paracetamol-induced liver damage (165 mg/kg body weight). Ethanolic leaf and fruit extracts were found to provide significant protection against paracetamol-induced hepatic necrosis. The leaf extracts show better hepatoprotective potential as compared to the fruit extracts.

### 6.3. Hypocholesterolemic Activity

Cheurfa et al. (2015) [27] studied the hypocholesterolemic activity of alcoholic and aqueous extracts of *P. lentiscus* leaves. Lipid parameters such as total cholesterol, triacylglyceride, low-density lipoprotein, very-low-density lipoprotein, and high-density lipoprotein were measured in the plasma. The administration of *P. lentiscus* extracts (200 mg/kg) produced in the ethanolic extract resulted in a significant decrease in total cholesterol, triacylglycerides, and low-density lipoprotein cholesterol (154 and 99 mg/dL, respectively). The aqueous extract produced a significant decrease in total cholesterol and triacylglycerides (203 and 97 mg/dL, respectively).

### 6.4. Anti-Ulcer Activity

Dellai et al. (2013) [71] compared the intensity of the anti-ulcer effects of lentisk leaf extracts prepared in different solvents. (chloroformic, ethyl acetate, methanolic, and aqueous extracts). The aqueous extract of *P. lentiscus* leaves showed an excellent protective effect against HCl/ethanol-induced gastric lesions in rats. The effect of lentisk leaf extract on behavioral, histological, and biochemical alterations caused by a metal neurotoxin (aluminum) in male albino mice was examined. The ethanolic extract and its isolated bioactive compounds were found to have a dose-dependent neuroprotective effect, with the highest effect obtained with myricetin rhamnoside (IC50 = 0.04 mM) [33].

## 7. Conclusions and Perspectives

Several phenolic acids were detected, among which gallic acid and galloyl derivatives were the major compounds. As for flavonoids, myricetin-3-O-rhamnoside was found to be the predominant flavonol glycoside and quercetin-3-glucoside was found to be the major flavonol in *P. lentiscus*. Flavanols assigned as procyanidin B1, procyanidin B2, epicatechin, catechin, epigallocatechin gallate, and epicatechin gallate were recovered. This review also covers investigations conducted on the biological potential of *P. lentiscus* phenolic extracts. The results gathered in these studies indicate that the *P. lentiscus* extracts have antioxidant, anti-inflammatory, anticancer, and antimicrobial properties in agreement with the traditional uses of this plant.

The observed anti-inflammatory effect is likely due to the presence of antioxidant compounds in the extract. There is a significant positive correlation between the antioxidant activity results and the amount of total phenols. The antioxidant capacity of *P. lentiscus* extract has been shown to be due primarily to gallic acids and their galloyl derivatives (5-Ogalloyl, 3,5-O-digalloyl, and 3,4,5-tri-O-galloyl). The anticancer potential could be due to flavonoids such as myricetin. The antimicrobial activity observed could be due to the presence of flavonoid compounds (kaempferol, myricetin, quercetin, and rutin) in the lentisk extracts.

To better analyze the correlation between the chemical composition and the pharmacological effect in relation to the traditional use of this plant, it would be interesting to use chiometric tools by applying metabolomics. Moreover, additional investigations on its pharmacological potential in vivo biological models are strongly recommended

These results should be supported by future studies regarding the possible toxic effects of lentisk extract so as to confirm the use of the plant in the treatment of various diseases.

*P. lentiscus* extracts remain a considerable source of natural antioxidants. This plant could be a promising source of natural antioxidants in food and pharmaceutical industries as alternatives to certain synthetic antioxidant additives.

This review can help in moving towards improving the production of phenolic molecules responsible for a given pharmacological potential by focusing on the various factors that can influence the composition and yield. It can therefore also help the potential development of a micro-industrial sector based on the valorization of lentisk polyphenols.

## Figures and Tables

**Figure 1 plants-12-00279-f001:**
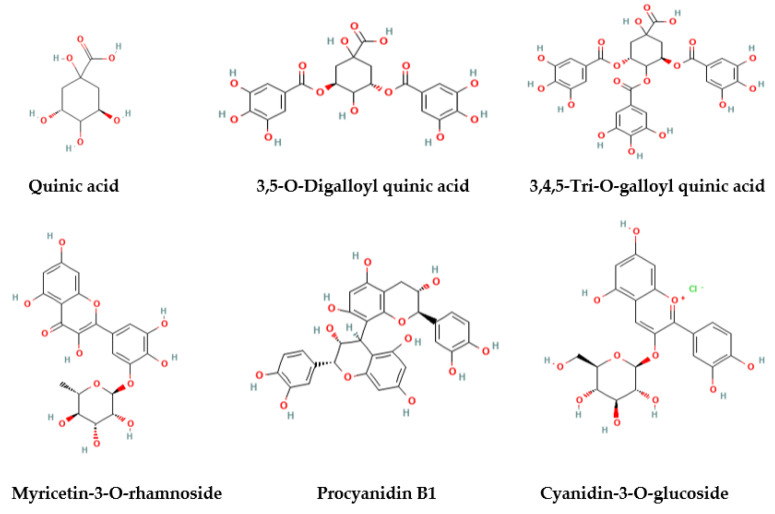
Structure of major phenolic acids and flavonoids detected in *P. lentiscus* extracts.

**Table 1 plants-12-00279-t001:** Lentisk phenolic classes.

Plant Part	Origin	Type of Extract	Classes Detected	Content			Analysis Method	Ref
Leaves	Algeria	Ethanolic extract	Total phenols	0.90 mg pyrocatechol Eq/g DM			FC (Folin–Ciocâlteu method) (Pyrocatechol as standard (Std), Eq: equivalent)	[23]
Leaves	Algeria	Aqueous extracts obtained from chloroform partition (solvent fractionation)	Total flavonoids Tannins	44.25 mg Q Eq/g dry matter (DM) 997.8 mg TA Eq/g DM			ATC (Aluminum Trichloride Calorimetric method: quercetin; Std) BSA (Bovine Serum Albumin method) (Tannic acid: Std)	[24]
Fruits	Algeria	Acetonic extract	Total phenols Total flavonoids	250.6 mg GA Eq/g DM 20.6 mg of Q Eq/g DM			FC (GA: gallic acid; Std) ATC (Quercetine (Q): Std)	[25]
Leaves	Algeria	Chloroform extract Methanolic extract Aqueous extract Ethyl acetate extract	Total phenols Total flavonoids	207 mg GA Eq/g DM 390 mg GA Eq/g DM 13 mg Q Eq/g DM 82.3 mg Q Eq/g DM			FC ATC	[26]
Leaves	Algeria	Aqueous extract (decoction) Ethanolic extract	Total flavonoids	3.1 mg Q Eq/g DM 8.2 mg Q Eq/g DM			ATC	[27]
leaves	Algeria	Aqueous extract (infusion)	Total phenols	119.7 mg GAE/g DM			FC	[28]
Leaves Stems Fruits Roots Leaves Stems Fruits Roots Leaves Stems Fruits Roots	Algeria Algeria Algeria	Methanolic extract Methanolic extract Methanolic extract	Total phenols Total flavonoids Tannins	216.2 mg GA Eq/g DM 121.4 mg GA Eq/g DM 103.3 mg GA Eq/g DM 30.1 mg GA Eq/g DM 19.1 mg C Eq/g DM 16.7 mg C Eq/g DM 4.7 mg C Eq/g DM 4.2 mg C Eq/g DM 121.5 mg C Eq/g DM 80.2 mg C Eq/g DM 7.8 mg C Eq/g DM 7.1 mg C Eq/g DM			FC ATC (Catechine: standard; Std) Vanillin method	[29]
Leaves	Algeria	Methanolic extract (maceration)	Total phenols Total flavonoids Flavonols Proanthocyanidins Total tannins	238.3 mg GA Eq/g DM 19.5 mg Q Eq/g DM 7.2 mg Q Eq/g DM 39.2 mg C Eq/g DM 100.6 mg TA Eq/g DM			FC ATC ATC and sodium acetate method Butanol–HCl method BSA (Tannic acid: Std)	[30]
Leaves	Algeria	Methanolic extract (maceration) Aqueous extract (decotion) Methanolic extract (maceration) Aqueous extract (decotion)	Total phenols Total flavonoids	286 mg GA Eq/g DM 227 mg GA Eq/g DM 15 mg Q Eq/g DM 11 mg Q Eq/g DM			FC ATC	[31]
Berries	Algeria	Ethanolic extract Solid–liquid “soxhlet” extraction	Total phenols Total flavonoids	955.2 mg GA Eq/g DM 13.4 mg Q Eq/g DM			FC ATC	[32]
Leaves	Algeria	Ethanolic extract	Total phenols Total flavonoids Myricetin-rhamnoside Quercetin- rhamnoside	95.8 mg GA Eq/g DM 5.1 mg Q Eq/g DM 3.6 mg/g DM 1.6 mg/g DM			FC ATC Preparative-HPLC purification ^1^H-NMR analysis	[33]
Leaves	Algeria	Ethyl acetate fraction from ethanolic extract	Total flavonoids	278.5 mg R Eq/g DM			ATC (Rutin: Std)	[34]
Aerial parts February May August	Greece	Methanolic extracts	Total phenols	483 mg GA Eq/g DM 588 mg GA Eq/g 5 DM 81 mg GA Eq/g DM			FC	[35]
Vegetable oil	Morocco		Total phenols Total flavonoids	122.7 mg GA Eq/g DM 18.4 mg Q Eq/g DM			FC ATC	[36]
Leaves Twigs Fruits	Morocco	Soxhlet extraction Solvants: [(hexane, ethyl acetate, methanol, ethanol) and water (infusion)]	Total phenols	67.8–345.9 mg GA Eq/g DM 48.7–302 mg GA Eq/g DM 36.4–192.6 mg GA Eq/g DM			FC	[37]
Femal and mal shrubs Leaves DP FS EF LF Stems DP FS EF LF Flowers FS Fruits (Female) EF LF	Tunisia	Aqueous acetone (80:20; *v*/*v*) extracts Maceration under magnetic stirring	TPC FVD TAN	TPC 193.8 125.8 160.4 96.8 49.6 22.9 43.4 17.4 55.0 59.4 27.1	FVD 47.5 28.1 40.9 24.4 7.8 2.6 6.7 1.7 28.1 9.1 4.5	TAN 68.4 57.3 31.8 30.7 41.6 33.8 30.7 20.4 26.6 22.1 11.6	TPC: total phenols (mg GA Eq/g DM) FC FVD: Total flavonoids (mg R Eq/g DM) Aluminum ATC TAN: Tannins (mg C Eq/g DM) Vanillin method	[38]

DP: Dormancy period; FS: flowering stage; EF: early fruiting stages; LF: late fruiting stage.

**Table 2 plants-12-00279-t002:** Individual phenolic compounds identified in lentisk extracts.

Compounds Detected	Plant Part	Type of Extract	Analysis Method	Provenance of the Plant	Ref
Catechin β-glucogallin Quercitrin gallate	Leaves	Methanolic extract	HPLC-ESI-QTOF-MS	Algeria	[3]
Glucogallin, gallic acid, galloyl quinic acid digalloyl quinic acid, trigalloyl quinic acid	Leaves	Ethanolic extract (maceration under stirrig)	UHPLC-MS	Algeria	[43]
Gallic acid Catechin Syringic acid Ellagic acid Quercetin 3-O-rhamnoside Luteolin	Leaves	Ethanolic extracts (maceration)	Fourier transform infrared spectroscopy HPLC-DAD	Algeria	[44]
Myricetin-3-O-glucuronide Myricetin-3-O-glucoside Myricetin-3-O-rhamnoside Quercetin-3-O-rhamnoside Catechine	Leaves	Hydro-methanolic extract (80%)	HPLC UV/Vis PDA	Croatia	[1]
Hydroxybenzoic acid derivatives; Hydroxycinnamic acid derivatives; Hydroxy cyclohexane carboxylic acid derivatives	Leaves	Ethanolic extracts	LC-ESI-MS/MS	Egypt	[45]
Quinic acid derivatives 3,5-O-digalloyl quinic acid 3,4,5-O-trigalloyl quinic acid Luteolin-3-O-rutinoside Epicatechin-3-gallate Quercetin-3-O-glucuronide 5,7-dihydroxyflavone	Leaves	Methanolic extract Solid-liquid “soxhlet” extraction	UHPLC-ESI-MS	Egypt	[46]
Kaempferol Myricetin Quercetin Luteolin Genistein Taxifolin	Leaves	Ethanolic extract	Reverse phase (HPLC)-UV/VIS array electrospray ionization (ESI)-mass spectrometry (MS)	Israel	[47]
Galloyl derivatives Catechin	Leaves	Hydro-ethanolic extract (7:3 *v*/*v*)	HPLC	Israel	[48]
Gallic acid and galloyl derivatives of glucose Galloyl derivatives of quinic acid Myricetin Quercetin glycosides: delphinidin 3-O-glucoside and cyanidin 3-O-glucoside	Leaves	Ethanolic extract	Semi-preparative HPLC HPLC-MS analysis with 1H- and 13C NMR.	Italy	[49]
Cyanidin 3-O-glucoside Delphinidin 3-O-glucoside Cyanidin 3-O-arabinoside	Fruits	Methanolic extract	HPLC-DAD mass spectrometry (MS)	Italy	[50]
Shikimic acid, quinic acid, acid derivatives, gallic acid, Glabrol, proanthocyanidin prodelphinidin B-4, (dimer of gallocatechin and epigallocathechin), catechin or epicatechin, epi)gallocatechin, (epi)catechin gallate, myricetin glycosides, myricetin derivatives, quercetin glycosides, quercetin O-galloylpentoside, kaempferol, luteolin	Leaves	Methanolic fraction -from chlorformic extract by sonication	LC-ESI-MS/MS	Italy	[51]
Myricetin glucopyranoside Myricetin-3-O-rutinoside Quercetin-3,4-diglucoside Quercetin-3-O-rutinoside Quercetin-3-O-galattoside Myricetin-3-O-rhamnoside Myricetin-3-O-xylopyranoside Quercetin-3-O-glucoside	Leaves/fruits	Aqueous extract (reflux heating)	LC-MS	Italy	[52]
Gallic acid, protocatechuic acid, p-hydroxybenzoic acid, vanillic acid isomer, caffeic acid, syringic acid isomer, p-coumaric acid, ferulic acid, rutin, quercetin 3-O-glucoside	Edible fixed oil from fruits	Hydro-methanolic extract (methanol 80%, *v*/*v*)	UHPLC-ESI-MS/MS and HPLC-DAD	Italy	[53]
Gallic acid, tyrosol, hydroxyphenylacetic acid, Vanillic acid, p-coumaric acid, methoxycinnamic acid, carnosic acid, salycilic acid, luteolin, kaempferol, naphtoresorcinol	Edible oil from fruits	Hydro-methanolic extract	HPLC-DAD/MSD	Tunisia	[54]
1.5 or 3.5 Dicaffeoylquinic acid Ellagic acid Catechin Luteolin-7-glucosid Isoquercetin Kaempferol rutinoside Rutin	Leaves	Aqueous extract (maceration uder magnetic agitation)	HPLC-MS	Tunisia	[55]

HPLC: High-performance liquid chromatography; HPLC-DAD-MS: high-performance liquid chromatography (HPLC)–diode array detection (DAD)–mass spectrometry (MS) analysis; HPLC-ESI-QTOF-MS: high-performance liquid chromatography with diode array coupled to electrospray ionization mass spectrometry; HPLC-DAD: high-performance liquid chromatography with diode array detection analysis; HPLC UV/Vis PDA: high-performance liquid chromatography (HPLC) coupled with UV/Vis PDA detector. ND: Not determined.

## Data Availability

Not applicable.

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
