# Peer review of "A Review of *Pistacia lentiscus* Polyphenols: Chemical Diversity and Pharmacological Activities"

_plants, 2023, doi:10.3390/plants12020279_

Round 1

Reviewer 1 Report

P. lentiscus L., a widely distributed plant in the Mediterranean countries, has been used as a remedy in traditional medicine for hypertension, inflammation, and gastrointestinal complaints treatment. Several compounds can be responsible for the effective biological activities of this plant. Other researches revealed the existence of flavonoids, anthocyanins, triterpenoids, tannins, and phenolic acids. The literature is diverse, encompassing a vast range of research studies. Therefore, I recommend major revision before the paper can be published in Plants.

My comments are listed below:

·       Abstract: Lines 24-27 need to be clarified; re-write them.

·       Introduction: There is a need to change the order of the second and first paragraphs.

·       This review is not revealed the search strategies, inclusion and exclusion criteria and risk of bias assessment for individual studies; therefore, there is a need to add a material and methods section.

·       There is a need to add a figure to show the overview of chemical composition.

·       The manuscript is too descriptive and monotonous and has much redundant information. For instance, a some part of the section 4 text overlaps with section 5. Another problem is the need for concision. Many statements need to deliver crucial scientific information, and extra should be removed.

·       The Tables are extensive. The relevant information from the Tables appears to be given in the text. They could be considered supplementary material or omitted from the paper altogether.

·       The conclusion is so exhaustive that there is a need to make it should be concise. 

Author Response

Dear reviewer 1,

A review of Pistacia lentiscus polyphenols: diversity of chemical composition and pharmacologicalactivities

ChabhaSEHAKI , Nathalie JULLIAN , Fadila AYATI , Farida FERNANE , Eric GONTIER 

Reviewer 1

  • Comment 1: Abstract: Lines 24-27 (previous version)/need to be clarified; re-write them.

Answer: The paragraph: ‘’Interesting pharmacological applications has been proven, particularly in terms of the plant's antioxidant, antimicrobial, and anti-inflammatory activities. A positive correlation between the pharmacological and phytochemical aspects of lentisk is proven. This review should allow the optimization of the extraction and isolation of phenolic molecules in sufficient quantities for specific pharmacological activities’’

is reworded by: ‘’ The biological and therapeutic potentials of pistacia extracts have been evaluated in terms of antioxidant, antimicrobial and anti-inflammatory activities. Most of these activities are related to the phenolic composition of this plant The content of this review will undoubtedly contribute to the choice of techniques for isolating the different bioactive molecules contained in the pistacia lentiscus’’

  • Comment 2: Introduction: There is a need to change the order of the second and first paragraphs.

Answer: the order of the second and first paragraphs is changed

  • Comment 3: This review is not revealed the search strategies, inclusion and exclusion criteria and risk of bias assessment for individual studies; therefore, there is a need to add a material and methods section.

Answer: In order to show an overview of the references consulted, which allowed us to extract and write all the data put forward in this ‘Review’, the material and methods section has been added in the paragraph under the title Research methodology

  • Comment 4: There is a need to add a figure to show the overview of chemical composition.

Answer: The figure is added (figure 1)

Comment 5: The manuscript is too descriptive and monotonous and has much redundant information. For instance, a some part of the section 4 text overlaps with section 5. Another problem is the need for concision. Many statements need to deliver crucial scientific information, and extra should be removed.

Answer: both sections 4 and 5 have been combined into one section “prominent pharmacological activities” and redundant information has been removed

  • 6- Comment 6: The Tables are extensive. The relevant information from the Tables appears to be given in the text. They could be considered supplementary material or omitted from the paper altogether.

Answer: unnecessary data within the different tables are eliminated. We have kept only the data carrying necessary information. The other information are completed in the body of the text without having to create redundancies

7-Comment 7: The conclusion is so exhaustive that there is a need to make it should be concise. 

Answer: the conclusion is made more precise

Reviewer 2 Report

The manuscript “A review of Pistacia lentiscus polyphenols: diversity of chemical composition and pharmacological activities” was submitted to Plants for publication.

Broad comments:

The idea for the present review article is good and the article gives a good overview on the topic. In addition, the section are well-defined and summarized appropriately. The major deficit of the review, however, is the arrangement of the tables. Here, the authors must invest a bit of work to reorganize and structure them in a better way. Firstly, because the table must give a clearer view and secondly, because as they arranged now they don’t support the conclusions.

Table 1:

This table is essential for the conclusion as the authors emphasize the variety in phenolic contents between the different geographical origins, extracts, etc. However, to be able to compare the quantitative results, all values must be standardized, i.e. by referencing them to a comparable unit. For plants, this unit in general is mg of compound(s) per g or kg dried plant material. Comparing the concentrations in different extracts is only interesting within the same sample, but not within different accessions. Furthermore, the first two columns should be exchanged and the plant organ listed before the origin of plant material, as the plant organ usually has a greater influence on the amount. Also think about a way to simplify the column with the analysis methods. The way it is now needs a lot of space and the information is redundant, as usually the same few method are used for the quantitation of phenolics. Please also reconsider the values of reference [33], as neither HPLC purification or 1H NMR analysis allow proper quantitation, unless qNMR is used.

Table 2:

For this table, the plant origin is less important and, in my opinion, can be omitted. In this table it would be more important to only list well-defined and unambiguous compounds, such as Myricetin 3-O-rhamnoside and but no quercetinglucoside if not specified where the glucose molecule is attached. Please also avoid any mentiones of hexosides, as this is also no defined sugar. I would also arrange the table in the way, that the compounds are listed in the first column, then the organ and the extract from which it derived. The amount can be omitted and references can be summarized if there are repeated reports of a compound. E.g., myricetin-3-O-rhamnoside, leaves, methanol, [xx,xy,xz]. Also think about drawing the reported structures in one or two figures. Thus, the reader will get a good image of the compounds and chemical structures always look good in a paper.

Table 3:

Also this table need extensive revision. Please do not give observation as every author will speak of pronounced or significant effects. This is part of the game. Please only depict results, best IC50 values. If these are not available also effects and measured concentrations in parenthesis will do.

Specific comments:

Title:      Please facilitate, e.g. “Pistacia lentiscus polyphenols: Chemical diversity and pharmacological activities”.

Family name:     Please do not use italics (only for the species name).

Species name:   Please use the whole name only when given the first time in the abstract and main text. Afterwards, use P. lentiscus.

There are several spelling and other mistakes, which need to be corrected, e.g. line 14 (setting of colons), line 64 (pharmacology), line 197 (extracted), line 408 (flavonoids), line 411 (Pistacia), line 445 (pharmacology), line 456 (review).

Line 447:              Please write only “significant” instead of “very significant”

Line 462:              Please write “additional investigations” instead of “other investigations”.

Author Response

Dear reviewer 2,

Reviewer 2

  • Comment 1:

Table 1:

This table is essential for the conclusion as the authors emphasize the variety in phenolic contents between the different geographical origins, extracts, etc. However, to be able to compare the quantitative results, (i) all values must be standardized, i.e. by referencing them to a comparable unit. For plants, this unit in general is mg of compound(s) per g or kg dried plant material. Comparing the concentrations in different extracts is only interesting within the same sample, but not within different accessions. Furthermore, (ii) the first two columns should be exchanged and the plant organ listed before the origin of plant material, as the plant organ usually has a greater influence on the amount. (iii) Also think about a way to simplify the column with the analysis methods. The way it is now needs a lot of space and the information is redundant, as usually the same few method are used for the quantitation of phenolics. (iv) Please also reconsider the values of reference [33], as neither HPLC purification nor 1H NMR analysis allows proper quantification, unless qNMR is used.

Answers:

  • All quantitative values were referenced to a comparable unit (mg compound/g dry matter)

  • The first two columns (geographical origin and plant material) have been inverted

  • We have simplified the presentation of the data shown in the analysis methods; using acronyms instead of the full name of each method. The full name of each method was only identified in its first appearance in the table.

  • Purification was performed by prep HPLC. The compounds were collected, rapidly evaporated and their content (quantification) was measured and expressed as mg per g dry weight. 1H NMR analysis was performed using the Bruker Ultrashield 600 MHz NMR spectrometer equipped with a cryogenic TXI probe head. The 1H NMR data were analyzed using Bruker Topspin software version 3.2.

Table 2:

For this table, (i) the plant origin is less important and, in my opinion, can be omitted.

(ii) In this table it would be more important to only list well-defined and unambiguous compounds, such as Myricetin 3-O-rhamnoside and but no quercetin glucoside if not specified where the glucose molecule is attached. Please also (iii) avoid any mentions of hexosides, as this is also no defined sugar. (iv) It would also arrange the table in the way,  that the compounds are listed in the first column, then the organ and the extract from which it derived. (v) The amount can be omitted and references can be summarized if there are repeated reports of a compound. E.g., myricetin-3-O-rhamnoside, leaves, methanol, [xx,xy,xz]. (vi) Also think about drawing the reported structures in one or two figures. Thus, the reader will get a good image of the compounds and chemical structures always look good in a paper.

Answers:

  • Indeed, the vegetable origin is less important, but at the same time we find that the fact of mentioning the origin of the plant gives a complete information about the extract. In order to keep this information we have reversed the geographical origin in the last column

  • Ambiguous compounds with undefined sugar molecule positions such as quercetin glucoside have been removed from Table 2

  • Compounds with the mention hexoside are omitted from the table

  • The table is arranged so that the compounds are listed in the first column, then the organ and the extract from which it is derived.

  • It can be noticed that the same compound exists in different types of extracts analyzed by different methods of analysis, so in order to avoid cluttering the columns (type of extract and method of analysis) we have summarized the majority compounds detected according to each study

  • Structure of major phenolic acids and flavonoids detected in lentiscus extracts were presented in Figure 1

Table 3:

Also this table needs extensive revision. (i) Please do not give observation as every author will speak of pronounced or significant effects. This is part of the game. Please only depict results, best IC50 values. If these are not available also effects and measured concentrations in parenthesis will do.

  • The term observations is replaced by the term significant result, in this column we have reduced the information by keeping only the significant values (IC50 and/or measured concentrations), particularly for the antioxidant activity.
  • Only the most remarkable activities have been kept in table 3. The other activities are mentioned in the text

  • Specific comments:
    • Title:      Please facilitate, e.g. “Pistacia lentiscus polyphenols: Chemical diversity and pharmacological activities”.

Answer: A review of Pistacia lentiscus polyphenols: diversity of chemical composition and pharmacological activities is replaced by A review of Pistacia lentiscus polyphenols: Chemical diversity and pharmacological activities

  • Family name:     Please do not use italics (only for the species name).

Answer: The italicized formatting has been cancelled for the family Anacardiaceae of the plant Pistacia lentiscus

2.3. Species name:   Please use the whole name only when given the first time in the abstract and main text. Afterwards, use P. lentiscus.

Answer: The full name of the plant is mentioned only when it is given for the first time in the abstract and the body of the text. Pistacia lentiscus is replaced by P. lentiscus in the rest of the text

  • There are several spelling and other mistakes, which need to be corrected, e.g. line 64/line 316 in corrected version (pharmacology), line 197/line 119 (extracted), line 408/line165 (flavonoids), line 41/line 232 (Pistacia), line 445/line 316 (pharmacology), line 456/line 532 (review).

Answer: The mistakes are corrected in the new version

  • Line 447:  Please write only “significant” instead of “very significant”

Answer: lines 112/137/303… Very significant is replaced by significant in the text

  • Line 462:              Please write “additional investigations” instead of“other investigations”

Answer: line 547: Other investigation is replaced by additional investigation

Reviewer 3 Report

1. Introduction it is well argued

2. Quantitative analyses of Pistacia lentiscus phenolic classes it is complex

3.  Variability in phenolic classes in Pistacia lentiscus:Variable levels of total phenolic contents, flavonoids, and condensed tannins were found in all parts of P. lentiscus, the use of different extraction solvents highlighted the fact that  the total phenols, total flavonoids, and total flavonols decreased in the following order according to the solvent extracts used: ethanol > water > methanol > ethylacetate > dichloromethane - well argued

4. Identification and quantification of individual phenolic compounds -  well argued

5. I suggest  checking the article in the plagiarism platform, being Review

6. I suggest revising the English language

Author Response

Dear reviewer 3,

  1. I suggest checking the article in the plagiarism platform, being Review

Answer: the article has been submitted to Compilatio Magister. No similarity exceeded 1% except for www.ncbi.nlm.nih.gov corresponding to citations of article references where it could punctually reach 3% (i.e. same titles cited in the reference list).

  1. I suggest revising the English language

Answer: The English language has been revised (English correction certificate attached)

English has been revised by the MDPI English editing service under the English-Editing-Certificate-54126

Round 2

Reviewer 1 Report

Most of the suggestions have been incorporated by the authors in the revised manuscript. Therefore, no issue with considering it for publication.

Reviewer 2 Report

The requested changes were conducted.